# Experimental Insights into the Interplay between Histone Modifiers and p53 in Regulating Gene Expression

**DOI:** 10.3390/ijms241311032

**Published:** 2023-07-03

**Authors:** Hyun-Min Kim, Xiaoyu Zheng, Ethan Lee

**Affiliations:** Division of Natural and Applied Sciences, Duke Kunshan University, Kunshan 215316, China

**Keywords:** histone modifications, p53, cancer, gene regulation, chromatin structure

## Abstract

Chromatin structure plays a fundamental role in regulating gene expression, with histone modifiers shaping the structure of chromatin by adding or removing chemical changes to histone proteins. The p53 transcription factor controls gene expression, binds target genes, and regulates their activity. While p53 has been extensively studied in cancer research, specifically in relation to fundamental cellular processes, including gene transcription, apoptosis, and cell cycle progression, its association with histone modifiers has received limited attention. This review explores the interplay between histone modifiers and p53 in regulating gene expression. We discuss how histone modifications can influence how p53 binds to target genes and how this interplay can be disrupted in cancer cells. This review provides insights into the complex mechanisms underlying gene regulation and their implications for potential cancer therapy.

## 1. Introduction

Chromatin is the complex of DNA, histone proteins, and other associated proteins that make up the structure of chromosomes within the nucleus of eukaryotic cells. It is a well-organized and dynamic structure that plays a fundamental role in the packaging and regulating of DNA. The chromatin structure can either promote or inhibit gene expression, depending on the specific modifications present on the histones [1,2,3,4]. Enzymes known as histone modifiers are responsible for adding or removing chemical changes in the histone proteins, thereby shaping the chromatin structure. These modifications, such as acetylation, methylation, phosphorylation, and ubiquitination, exhibit diverse effects on chromatin structure and gene expression.

p53, as a transcription factor, plays a crucial role in controlling the expression of the various genes involved in cell cycle regulation, DNA repair, apoptosis, and other cellular processes [5,6,7]. It acts as a tumor suppressor by promoting cell cycle arrest or inducing apoptosis in response to DNA damage or cellular stress. Thus, p53 is one of the most crucial tumor suppressor genes in tumorigenesis. The activity of p53 is regulated by various mechanisms, including the post-translational modifications of both p53 and the histones surrounding it. 

The interplay between histone modifiers, such as histone acetyltransferases/deacetylase and histone methyltransferases/demethylase, and p53 is crucial for regulating gene expression, maintaining genomic stability, facilitating chromatin remodeling, and promoting DNA repair. The synergy between the two ensures proper cellular responses to DNA damage, stress signals, and other regulatory cues [8,9,10,11,12,13,14,15,16]. p53 is the guardian of the genome, whose activity is implicated in most types of cancers, so its post-transcriptional modifications have been extensively studied [16,17,18]. However, the connection between histone modifiers and p53 gained less attention despite the growing body of research on the involvement of histone modifiers in cancer. Therefore, this review primarily centers on exploring the impact of histone modifiers on p53 and their significance in regulating gene expression.

We implemented a systematic methodology to ensure a thorough literature review. Our approach involved searching the PubMed database using keywords pertinent to our research topic. Additionally, we scrutinized the reference lists of relevant articles to uncover any additional studies of relevance. To mitigate potential bias, we strived to encompass papers published in peer-reviewed journals until May 2023. 

## 2. The Interplay between Histone Modifiers and p53

Multiple mechanisms contribute to the interplay between histone modifiers and p53. Firstly, histone modifiers indirectly influence p53 function by modifying the structure and accessibility of p53 target genes within the chromatin (Figure 1). By establishing either an open or closed chromatin state, they impact p53’s ability to bind to specific DNA sequences and regulate gene expression [19,20]. Alternatively, p53 can directly interact with specific histone modifiers or their associated proteins, facilitating their recruitment to target genes. As a result, histone modifications near p53-binding sites can either enhance or suppress gene expression [15,21,22]. Additionally, p53 plays a role in regulating the expression of histone modifiers by binding to their promoters. Importantly, these mechanisms can occur simultaneously [23].

### 2.1. Histone Modifiers on p53

Packaging DNA into chromatin affects gene expression by making specific genes accessible to transcription factors, which are proteins that bind to DNA and control gene expression. Histone modifications affect chromatin structure by altering the interactions between histones and DNA. Each of these modifications can affect chromatin structure and gene expression differently. Modifications that are associated with active transcription, such as the acetylation of histone three and histone four (H3 and H4) or the di- or trimethylation (me) of H3K4, are commonly referred to as euchromatin modifications [27] (Figure 1A and Table 1). Conversely, modifications localized to inactive genes or regions, such as H3K9me and H3K27me, are often termed heterochromatin modifications.

The enzymes responsible for regulating post-translational epigenetic modifications on histones have been categorized into four groups based on their roles: writers, erasers, readers, and movers (Table 1) [28]. Writers add changes to histones and include DNA methyltransferases (DNMTs), histone lysine methyltransferases (KMTs), and histone acetyltransferases (HATs). These modifications can affect the chromatin structure and gene expression by either promoting or repressing gene transcription. Therefore, they play a crucial role in establishing and maintaining the epigenetic marks on histones, which modulates gene expression. Conversely, erasers remove post-translational modifications, including histone lysine demethylases (KDMs) and histone deacetylases (HDACs). By removing these modifications, erasers can reverse the effects of writers and restore the original state of histones. This dynamic regulation allows for the fine-tuning of gene expression and the potential for epigenetic remodeling. The readers describe bromodomain and chromodomain proteins that can “read” acetylated or methylated residues. Readers play a crucial role in translating the histone modifications into functional outcomes by facilitating the recruitment of effector proteins to specific genomic loci. Movers can remodel chromatin by moving nucleosomes, thus influencing gene transcription. By repositioning or rearranging nucleosomes, movers can modulate the accessibility of DNA to transcription factors and other regulatory proteins.

Furthermore, histone modifications can be classified into major groups based on the type of modification and the amino acid residue being modified, including acetylation, methylation, phosphorylation, ubiquitination, sumoylation, and ADP-ribosylation (Table 1). Among these modifications, methylation, acetylation, and phosphorylation are the primary ones observed in the interplay between histones and p53 (Figure 1). Building upon this understanding, this review primarily centers on exploring the impact of histone modifiers on p53 and their significance in regulating gene expression.

**Table 1 ijms-24-11032-t001:** Histone modifiers in chromatin. Histone modifying enzymes and their roles in histone modification.

**Chromatin Modification**	**Associated Transcription State**	**References**	
Acetylation of H3	Euchromatin/Active	[27]	
Acetylation of H4	Euchromatin/Active	[27]	
Di-/Trimethylation of H3K4	Euchromatin/Active	[29,30]	
Di-/Trimethylation of H3K9	Heterochromatin/Inactive	[29,30]	
Di-/Trimethylation of H3K27	Heterochromatin/Inactive	[31,32]	
**Enzyme Category**	**Enzyme Types**	**Description**	**Reference**
Writers	DNA methyltransferases (DNMTs)	Enzymes responsible for adding DNA methylation	[28]
	Histone lysine methyltransferases (KMTs)	Enzymes responsible for adding histone lysine methylation	
	Histone acetyltransferases (HATs)	Enzymes responsible for adding histone acetylation	
Erasers	Histone lysine demethylases (KDMs)	Enzymes responsible for removing histone lysine methylation	
	Histone deacetylases (HDACs)	Enzymes responsible for removing histone acetylation	
Readers	Bromodomain proteins	Proteins that can “read” acetylated residues on histones	
	Chromodomain proteins	Proteins that can “read” methylated residues on histones	
Movers	Nucleosome remodeling complexes	Enzymes that can move nucleosomes, aiding gene transcription	
**Histone Modifications**	**Amino Acid Residue**		
Acetylation	Lysine (K)		
Methylation	Lysine (K), Arginine (R)		
Phosphorylation	Serine (S), Threonine (T), Tyrosine (Y)		
Ubiquitination	Lysine (K)		
Sumoylation	Lysine (K)		
ADP-ribosylation	Various (E, D, R, C, K, and more)		

#### 2.1.1. Methylation

The primary amino acids that are susceptible to methylation are arginine and lysine (Figure 1B). Arginine methylation involves the monomethylation, asymmetric dimethylation, or symmetric dimethylation of arginine residues, which are mediated by three types of protein arginine methyltransferases (PRMT type I, II, III), and this modification plays a role in modulating protein function and various cellular activities. On the other hand, lysine methylation leads to the formation of mono-, di-, or trimethylated lysine residues. In histones, lysine methylation is a crucial epigenetic marker that influences gene expression by affecting chromatin structure and attracting proteins involved in transcriptional regulation. Lysine methylation can also occur on non-histone proteins, governing cellular processes such as DNA repair, signal transduction, and protein–protein interactions.

##### SET7/9

SET7/9, encoded by the SETD7 gene, was independently discovered in 2001 by Reinberg’s lab (named SET9) and Zhang’s lab (named SET7) [33,34,35]. Initially recognized as a methyltransferase involved in the methylation of H3K4, SET7/9 facilitates transcriptional activation by displacing the histone deacetylase NuRD complex (HDAC) [34]. Moreover, the observation that SET7/9-mediated H3K4 methylation enhances histone acetylation, which is associated with gene activation, implies that SET7/9 positively regulates transcription.

SET7/9 was also reported to methylate non-histone proteins, including p53 [19,35], and the SET7/9-mediated lysine methylation of p53 has been found to contribute to p53 activation [19]. SET7/9 methylate p53 is a transcription factor that regulates p21 expression, thereby enhancing p21 gene expression. Specifically, SET7/9 methylates p53 at K372 within the carboxyl–terminus regulatory region. This p53-K372 resulted in stabilizing a chromatin-bound fraction of p53. This, in turn, boosts the expression of p21, a target gene of p53, and promotes an increase in p53-mediated apoptosis. The methylation event on p53 also enhances the binding affinity between p21 and CDKs, resulting in the inhibition of CDK activity and the subsequent arrest of the cell cycle (Figure 2) [13,20].

Ivanov et al. used chromatin immunoprecipitation (ChIP) assays to analyze the levels of p53 binding and histone H4 acetylation at the promoter of the p21 gene [20]. Methylation of p53, by SET7/9, increased the binding of p53 to the promoter region of the p21 gene, resulting in the increased acetylation of histone H4 and the subsequent transcriptional activation and cell cycle arrest (Figure 2).

Interestingly, SET7/9-mediated methylation of p53 is required for the binding of acetyltransferase Tip60 to p53 and the subsequent acetylation of p53, and the lack of SET7/9 expression leads to defective cell cycle arrest upon DNA damage in mice, suggesting the coordination of two different modifications in p53 activation. However, the loss of SET7/9 is not as sufficient in contrast to p53^-/-^ mice [36].

A similar interplay between p53 and lysine methyltransferase has already been described [20]. Pre-methylation at K372-p53 enhances the subsequent acetylation of p53 by p300 upon DNA damage. However, pre-acetylation of the p53 inhibits the subsequent methylation of K372 by SET7/9 (SET7 and SET9)**,** suggesting that methylation must precede acetylation for a positive interplay between methylation and acetylation. Of note, several lysine residues near the site where SET7/9 methylates p53 can also be acetylated by CBP/p300 [20]. These lysine residues are numbered K370, K373, K381, and K382 in vitro and K373 and K382 in vivo. These results suggest that an interplay between histone modifiers and p53 is required for subsequent transcriptional activation. However, the exact mechanism of methylation-dependent acetylation of p53 remains to be explored.

##### G9a

G9a, a Set domain-containing protein, serves as the major histone lysine methyltransferase. It methylates histone H3 at both H3K9me1 and H3K9me2 modifications, earning its name G9a [37]. By adding methyl groups to histones, the G9a protein establishes and maintains epigenetic marks that regulate gene activity. It engages in various cellular processes, including embryonic development, cell differentiation, and the maintenance of cell identity.

Human G9a (hG9a) can regulate the expression of p21 in a manner that is independent of p53 and its methylation activity [21]. G9a positively regulates p21 expression independently of p53 and its histone methyltransferase activity. Oh et al. demonstrated that hG9a upregulates p21 via interaction with PCAF, and this activating complex is recruited to the p21 promoter upon DNA damage-inducing agent etoposide treatment. Ultimately, p21 induction by G9a inhibits cellular proliferation and leads to apoptosis in p53-null cells. This regulatory mechanism does not rely on the histone–lysine methyltransferase activity of G9a and functions through a pathway separate from p53 (Figure 3).

Similarly, hG9a stimulates p53’s activity independently of methylation by interacting with histone acetyltransferase CBP/p300, resulting in increased histone acetylation at the promoter of pro-apoptotic genes, including PUMA, thus inducing p53 transcriptional activity [22].

On the contrary, the mouse one (mG9a) blunted P53-dependent transcription in a methylation-specific manner (Figure 3). The differences in the regulation of P53 by hG9a and mG9a may be due to splicing variants. The human G9a (EHMT2) gene is present in cells as two splice variants (hG9a long and hG9a short), while mG9a is the product of the NG36–G9a transcript, which is similar to hG9a based on the amino acid sequences. The findings from two independent studies have identified that human G9a (hG9a) functions in a manner that is independent of methylation [21,22]. However, the specific mechanisms by which hG9a targets p53 and p21, either separately or in a coordinated manner, remain unclear. Further research is needed to elucidate the precise mechanisms by which hG9a modulates the expression of p53 and p21 and determine whether these regulations occur independently or through interconnected pathways.

##### PRMTs (Protein Arginine Methyltransferases)

PRMTs have crucial roles in various cellular processes, including transcriptional regulation, chromatin regulation, signal transduction, and DNA damage repair. They catalyze the transfer of a methyl group from S-adenosylmethionine to the guanidine nitrogen of arginine residues in proteins [38,39]. PRMT5 specifically methylates histone H4 at arginine 3 (H4R3), indirectly influencing p53 activity by affecting the transcriptional regulation of p53 target genes. Following DNA damage, PRMT5 methylates p53 at arginine residues R333, R335, and R337 [40]. Bypassing p53 through arginine methylation leads to apoptosis evasion and facilitates tumor growth [15] (Figure 4). Consistently, the depletion of PRMT5 triggers p53-mediated apoptosis, indicating that arginine methylation plays a role in controlling p53 activity. R337H mutation, prevalent in pediatric adrenocortical tumors in southern Brazil, also underscores the significance of arginine methylation in regulating p53-mediated events and oncogenesis [41,42].

In addition to the methylation of p53, PRMT5 has been shown to regulate chromatin structure and gene expression through its interaction with histones. By controlling the alternative splicing of crucial histone-modifying enzymes, such as TIP60 and KMT5C, PRMT5 can affect chromatin structure and, ultimately, impact DNA repair [43] (Figure 4). Therefore, PRMT5-mediated changes in histone methylation may indirectly affect p53 function by altering gene expression patterns.

##### JMJD2

While p53 is a target for epigenetic modulators, it can also target histone modifiers. The Jumonji C domain, containing the histone demethylase 2 (JMJD2) family of proteins, selectively demethylates H3K9me3 and H3K36me3. JMJD2B/KDM4B is a p53-inducible gene in response to DNA damage (Figure 5). p53 regulates JMJD2B gene expression by binding to a p53-consensus motif in the JMJD2B promoter. JMJD2B induction attenuates the transcription of key p53 transcriptional targets, including p21, PIG3, and PUMA, while silencing enhances the induction of the two [23]. JMJD2B-mediated histone demethylation is also critical for p53-mediated autophagy and survival in Nutlin-treated cancer cells [14]. 

##### EZH2

EZH2, also known as Enhancer of Zeste Homolog 2, is a vital protein involved in epigenetic regulation. It belongs to the Polycomb group protein family and serves as the catalytic subunit of the Polycomb Repressive Complex 2 (PRC2) [44]. Functioning as a methyltransferase, EZH2 adds methyl groups, specifically, to lysine 27 of histone H3 (H3K27) through its histone methyltransferase activity. This enzymatic function enables EZH2 to modify chromatin structure by depositing the repressive histone mark H3K27me3. The addition of methyl groups by EZH2 plays a pivotal role in gene silencing and epigenetic regulation.

Several studies provided valuable insights into the dynamic relationship between p53 and EZH2. Tang et al. demonstrated that activated p53 downregulates EZH2 gene expression by repressing the EZH2 gene promoter [45]. Additionally, their findings revealed that reducing EZH2 expression leads to impaired cell proliferation and G2/M arrest. These observations suggest that p53 controls the G2/M checkpoint by suppressing EZH2 expression. Yuan et al. also uncovered an intriguing interplay between Ezh2 and p53 in regulating inflammasome activation (Figure 6) [46]. Ezh2 competes with p53 for binding to the promoter of the lncRNA Neat1 gene. This competition allows Ezh2 to maintain the enrichment of H3K27 acetylation (H3K27ac) and chromatin accessibility, facilitating the transcription of Neat1 by p65. Consequently, inflammasome activation is promoted [46,47].

#### 2.1.2. Phosphorylation

Phosphorylation is a common post-translational modification that involves adding a phosphate group (PO_4_^3−^) to specific amino acid residues in proteins—typically serine, threonine, or tyrosine (Figure 1). This modification is catalyzed by protein kinases, which transfer the phosphate group from ATP to the target residue. Phosphorylation of p53 can occur at multiple sites in response to various stress signals. Phosphorylation of p53 at Ser15, in response to ionizing radiation, enhances the transcriptional activity of p53 by increasing its affinity for DNA to recruit coactivators such as CBP/p300 (Figure 7) [48,49].

Similarly, other studies reported that Ser20 undergoes phosphorylation following exposure to ionizing radiation, which could potentially weaken the binding of p53 to Mdm-2 to save p53 from ubiquitin-mediated degradation [50,51,52]. The phosphorylation of human p53 Ser-392 in the C-terminal regulatory domain also occurs following UV but not γ-irradiation [53,54], and it results in the enhancement of sequence-specific binding activity in vitro [55], possibly by promoting the stable tetramer form of p53 (Figure 7) [56]. These observations explain the activation of p53-regulated genes following DNA damage.

##### MAP Kinase Cascade

MAP kinase cascade is one of the major UV response pathways [57]. This pathway has three distinct components in mammalian cells: extracellular signal-regulated protein kinases (ERKs), p38 kinases, and stress-activated c-Jun N-terminal kinases (JNKs). These kinases participate in the regulation of cell proliferation, differentiation, stress responses, and apoptosis.

##### p38 MAP Kinase

p38 can directly phosphorylate and activate p53. Upon activation, p38 phosphorylates specific serine residues on p53, such as Ser15 and Ser392 [57,58,59], leading to increased p53 stability, transcriptional activity, and the subsequent induction of downstream target genes involved in cell cycle arrest, DNA repair, and apoptosis (Figure 8). Ser15 phosphorylation also stabilizes p53 by reducing its interaction with MDM2, a negative regulatory partner [60]. Hence, phosphorylation of p53 is likely to play an essential role in regulating its activity.

p38 can also indirectly influence histone modifications through various mechanisms. p38 phosphorylates and activates downstream targets, including kinases and transcription factors, which can, in turn, modulate histone modifications. For example, MSK1, a downstream target of the MAPK pathway, can be activated by p38 MAPK [61]. Upon activation, MSK1/2 can phosphorylate specific residues of histone H3, leading to the modification of chromatin structure and the regulation of gene expression (Figure 8). Specifically, it has been demonstrated to phosphorylate histone H3 at serine 10 (H3S10) and serine 28 (H3S28) residues [62,63].

When activated by the p38 MAPK pathway, MSK1 interacts with p53 and is recruited to the p21 promoter, where it phosphorylates histone H3 in a p53-dependent manner. Therefore, MSK1 plays a role in activating the expression of the p21 gene [64]. This enhances the transcriptional activation of p21, as evidenced by in vitro chromatin transcription and cell-based analyses. Overall, p38 MAPK activates p53 and indirectly influences histone modifications, while histone modifications can modulate p53 function. These interconnected relationships contribute to the intricate regulatory networks involved in cellular stress responses, DNA damage repair, and gene expression control.

##### RSK2

RSK2 is a p90 ribosomal S6 kinase family member that is activated by growth factors, peptide hormones, and neurotransmitters via MAPK/ERK signaling (ERK1 and ERK2). It is critical in regulating gene transcription by phosphorylating CBP at Ser133 [65]. Additionally, RSK2 has been reported to phosphorylate both histone H3 and p53 [66]. When cells are stimulated with UV or EGF, RSK2 is activated through the MAPK cascade, and it phosphorylates p53 protein at Ser15 (Figure 9) [66]. Authors further proposed that the RSK2–p53 complex then translocated to the nucleus, where RSK2 phosphorylates histone H3 at Ser10 and induces expression of target genes. These findings suggest that the interplay of RSK2–p53–histone H3 may contribute to transcriptional regulation, chromatin remodeling, and cell cycle regulation.

#### 2.1.3. Acetylation

Acetylation plays a significant role in modulating the transcriptional activity of p53, serving as a substantial modification (Figure 1) [67,68,69]. It is a post-translational modification that adds an acetyl group (-COCH3) to specific amino acid residues in proteins, predominantly lysine. This modification is catalyzed by enzymes known as histone acetyltransferases (HATs) or lysine acetyltransferases (KATs) [70]. In response to DNA damage, acetylation dynamics play a role in chromatin regulation. In the context of cancer, the pathways responsible for acetylation are subject to mutations or abnormal expression [71].

##### CBP/p300

CREB-binding protein (CBP/p300), in mediating p53 acetylation and its consequential effect on p53 activity, has been illuminated in previous studies [72,73,74,75]. Extensive research has delved into unraveling the impact of acetylation on the regulation of p53’s functionality. The interaction between CBP/p300 and p53 leads to the acetylation of specific lysine residues within the regulatory part of p53, resulting in a conformational change that enhances its DNA binding activity [76].

The ability of p53 to be acetylated was subsequently confirmed using acetylation-specific antibodies [77]. Sakaguchi et al. show that CBP/p300 acetylates K382 of p53 using a polyclonal antiserum, specific for p53, that is phosphorylated or acetylated at specific residues, while ATM phosphorylates S33 and S37 in response to UV irradiation. The acetylated p53 leads to increased binding to DNA. After DNA damage from irradiation, acetylation occurs at specific lysine residues, K382 and K320 of the p53, resulting in the recruitment of coactivators, such as CBP/p300 and TRRAP, to the p21 promoter and increasing histone acetylation. This suggests that a cascade of acetylation, in which p53-dependent recruitment of coactivators/HATs occurs, is essential for p53 to function correctly (Figure 10) [78].

##### SIRT1

SIRT1 was the first enzyme identified to target p53, which is not a histone, for deacetylation [79]. SIRT1 plays a role in regulating the cellular response to DNA damage by modifying the activity of p53 through the removal of acetyl groups from lysine residues [80,81,82,83,84]. Specifically, the deacetylation of K382 by SIRT1 inhibits the ability of p53 to activate transcription. This process leads to the degradation of p53, resulting in reduced apoptosis and increased cell survival when faced with DNA damage (Figure 10) [8,81]. In contrast, SIRT1 also promotes apoptosis in mouse embryonic stem cells through a transcription-independent mechanism involving p53. SIRT1 deacetylates p53 at K379 (equivalent to human K382) and prevents its nuclear translocation. Consequently, p53 translocates to the mitochondrial outer membrane and releases the pro-apoptotic protein BAX, as reviewed by [8].

##### TIP60

Tip60 is another histone acetylase linked to DNA damage repair and apoptosis [85,86,87,88,89,90,91]. There were two research groups who found that Tip60 induces K120 acetylation in the DNA binding domain upon DNA damage [86,92]. Lysine 120 (K120) acetylation occurs rapidly after DNA damage, and it is catalyzed by the MYST histone acetyltransferases hMOF and TIP60 (Figure 11) [92]. The mutation of K120 to arginine (K120R) debilitates K120 acetylation and blocks the transcription of pro-apoptotic target genes, such as BAX and PUMA, which, in turn, diminishes p53-mediated apoptosis without affecting cell cycle arrest. Additionally, the acetyl-K120 of p53 specifically accumulates at pro-apoptotic target genes. Additionally, studies indicate that Tip60–TRRAP complexes relocated to gamma-H2AX foci in response to DNA damage [88,93] and are crucial for the apoptotic response [89].

K120 mutation, found in human cancers, further suggests that defective K120 acetylation may contribute to tumorigenesis [92]. Likewise, tumor-associated K120R mutation abrogated p53-dependent apoptosis, suggesting that p53 activity was blocked in human cancer with the same mutations [86]. As such, the relationship between Tip60-mediated p53 acetylation and the consequent induction of apoptosis is a prominent issue that warrants further investigation.

Li et al. generated mutant mice with lysine to arginine mutations at one (K117R, K120 in humans) or three (3KR; K117R + K161R + K162R) sites in the p53 [94]. The results showed that K117R cells could still cause cell cycle arrest and senescence but not apoptosis, while 3KR cells failed to perform any of these processes, indicating that a fine tune of acetylation modulates downstream of the DNA damage repair pathway. Consistently, while acetylation at K120 enhances apoptosis induction, acetylation at K164 promotes cell cycle arrest, suggesting that acetylation of the two lysine residues helps distinguish the cell cycle arrest and apoptotic functions of p53 [94,95]. This highlights the importance of the interaction between histone acetyltransferases and p53 in regulating various cellular processes. Further studies are needed to understand the molecular mechanisms underlying this complex interplay fully.

The above-mentioned results indicate that histone modifications are vital for regulating p53 function, with specific enzymes responsible for acetylation and lysine methylation playing a role in activating and stabilizing p53. These findings significantly impact our comprehension of p53 and its involvement in cancer, as aberrant p53 activity is frequently observed in various cancer types.

## 3. Perspectives

The relationship between p53 and histone modifiers is intricate and bidirectional, with both factors capable of modifying each other. Upon DNA damage or cellular stress, p53 recruits histone modifiers to specific genomic loci, influencing chromatin structure and gene expression. Conversely, histone modifiers can impact p53 activity by altering its post-translational modifications or how it binds to DNA. This interplay between p53 and histone modifiers is crucial for regulating gene expression, maintaining genomic stability, facilitating chromatin remodeling, and supporting DNA repair. It ensures appropriate cellular responses to DNA damage, stress signals, and regulatory cues, although their interaction varies depending on the specific cellular environment or stimulus.

Dysregulation of histone modifiers and p53 pathways can contribute to treatment resistance in cancer cells. This involvement of histone modifiers and p53 in treatment resistance has prompted the development of multiple epigenetic anticancer drugs approved by regulatory authorities. Epigenetic therapies have emerged as a promising avenue for cancer treatment. Notably, the U.S. Food and Drug Administration (FDA) has granted approval for drugs targeting diverse epigenetic categories, including HDAC, DNMT, and EZH2 (Appendix A). HDAC inhibitors can modulate gene expression by increasing histone acetylation levels and promoting the activation of tumor suppressor genes [96,97,98,99]. Acetylation inhibitors can activate tumor suppressor transcription. By inhibiting the activity of histone deacetylases (HDACs), acetylation inhibitors increase histone acetylation levels, leading to a more open chromatin structure. This chromatin remodeling allows easier access of transcription factors, including those involved in tumor suppressor pathways, to the target DNA sequences. With the increased accessibility, tumor suppressor genes, such as p53, can be effectively transcribed and translated into functional proteins. Therefore, the activation of tumor suppressor transcription by acetylation inhibitors contributes to suppressing tumor progression.

Combination therapies in cancer treatment involve utilizing multiple drugs or treatment approaches, simultaneously, to enhance the effectiveness of treatment. These therapies have the potential to overcome drug resistance and improve treatment outcomes for individuals with cancer [100,101]. By investigating the molecular mechanisms contributing to treatment resistance, researchers can develop innovative combination therapies targeting histone modifiers and p53. In the specific case of histone modifiers and p53, the proposed combination therapies would incorporate drugs that target both histone modifiers (such as histone deacetylase inhibitors) and p53 (such as p53 activators or stabilizers). This approach aims to address treatment resistance mechanisms. Through synergistic effects, this combination therapy has the potential to enhance treatment efficacy and overcome the limitations associated with using single-agent therapies, ultimately leading to improved overall responses and outcomes for cancer patients.

Despite the extensive individual research on histone modifiers and p53, their interplay has received relatively less attention. However, the relationship between p53 and histone modifiers plays a critical role in gene expression regulation, genomic stability, chromatin remodeling, and DNA repair. Understanding the intricate mechanisms of p53 and histone modifiers is highly relevant in cancer therapy, especially considering the frequent occurrence of p53 mutations and dysregulation. Overall, comprehending the interplay between p53 and histone modifiers is crucial for understanding treatment resistance and holds promise for developing innovative cancer therapies that can effectively target and modulate these critical factors.

## Figures and Tables

**Figure 1 ijms-24-11032-f001:**
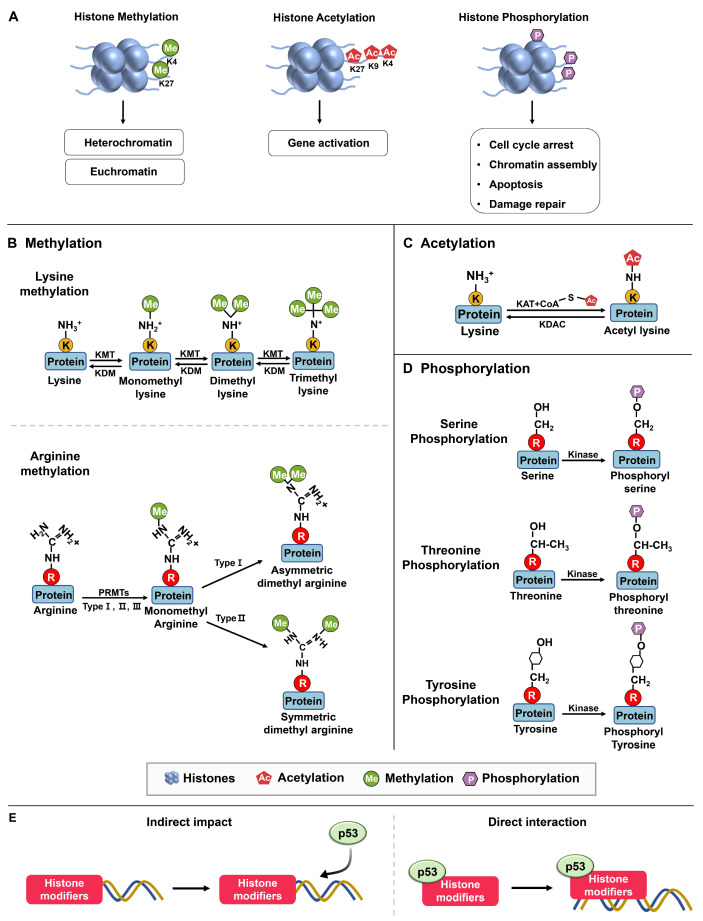
Histone modifications. (**A**) Methylation and demethylation are catalyzed by histone methyltransferase and histone demethylase, respectively. Euchromatin is characterized by specific molecular marks that indicate active gene expression, including histone acetylation, H3K4 trimethylation (H3K4me3), and H3K36 trimethylation (H3K36me3). In contrast, heterochromatin is marked by different modifications associated with gene repression and chromatin compaction. These marks include H3K9 trimethylation (H3K9me3) and H3K27 trimethylation (H3K27me3). Acetylation occurs in lysine residues catalyzed by histone acetyltransferase, while deacetylation is catalyzed by histone acetyltransferase. Phosphorylation: Kinases and phosphatases are enzymes involved in the addition and removal of phosphate groups, respectively, on proteins [24]. (**B**) Protein methylation occurs on lysine and arginine residues in histone and non-histone proteins through protein methyltransferases. The specific methyltransferases and demethylases reversibly regulated lysine methylation and demethylation from mono to trimethylation. Arginine methylations are induced in three types, including monomethylation, asymmetric dimethylation, and symmetric dimethylation [25]. (**C**) Acetyltransferases (KATs) transfer the acetyl group from acetyl–CoA to specific lysine residues in proteins, while acetylation can be reversed by lysine deacetylases (KDACs). (**D**) Protein phosphorylation occurs in serine, threonine, and tyrosine residues [26]. (**E**) Histone modifiers and p53 interact through mechanisms: Indirect chromatin modifications and direct recruitment target genes, affecting gene expression.

**Figure 2 ijms-24-11032-f002:**
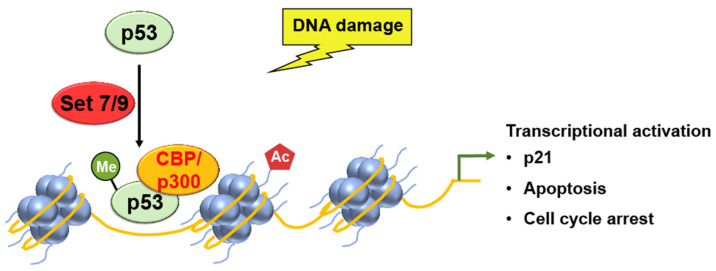
SET7/9 methylation activates p53, which leads to the transcriptional activation of p21 gene expression, as well as p53-mediated apoptosis and cell cycle arrest.

**Figure 3 ijms-24-11032-f003:**
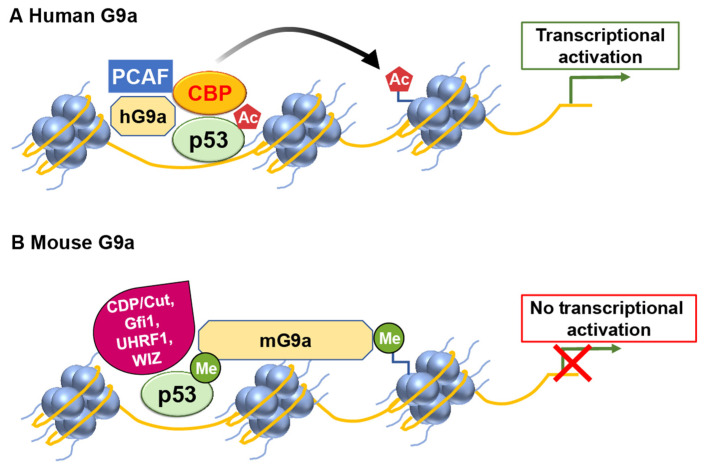
G9a has the potential to modulate p53 transcriptional activity in a differential manner. (**A**) In humans, G9a functions as a coactivator for p53 by recruiting histone acetyltransferases (HATs) such as CBP and PCAF. (**B**) However, mouse G9a exerts a repressive effect on p53 transcriptional activity.

**Figure 4 ijms-24-11032-f004:**
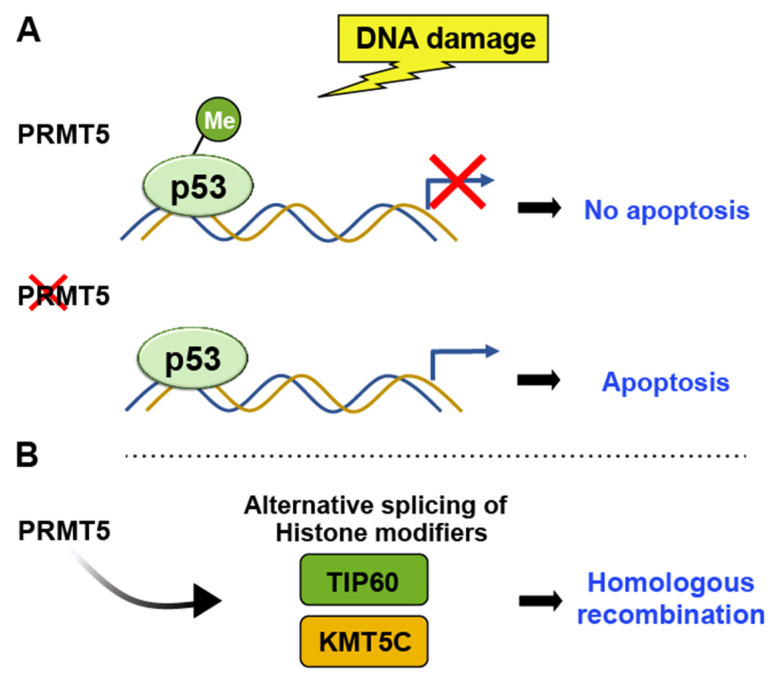
PRMT5 methylates H4R3, indirectly influencing p53 activity by affecting the transcriptional regulation of p53 target genes. (**A**) Arginine methylation-induced bypassing of p53 leads to the evasion of apoptosis and facilitates tumor growth, whereas the depletion of PRMT5 induces p53-mediated apoptosis. (**B**) PRMT5 controls the alternative splicing of key histone-modifying enzymes such as TIP60 and KMT5C, thereby influencing chromatin structure and influencing DNA repair pathway.

**Figure 5 ijms-24-11032-f005:**
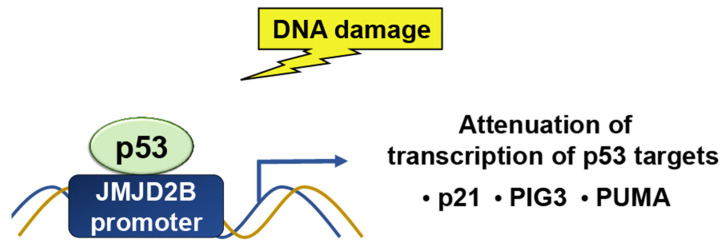
In response to DNA damage, p53 binds to a p53-consensus motif in the JMJD2B promoter; hence, p53 controls the expression of the JMJD2B gene. The induction of JMJD2B, in turn, suppresses the transcription of important p53 targets, such as p21, PIG3, and PUMA.

**Figure 6 ijms-24-11032-f006:**
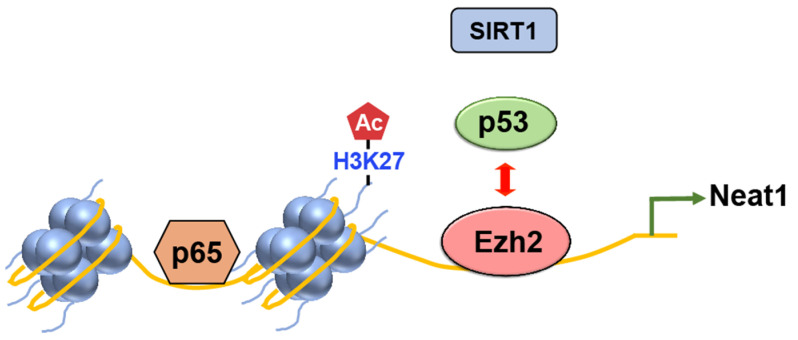
The competition between Ezh2 and p53 regulates inflammasome activation in mice. Upon exposure to inflammasome inducers, Ezh2 inhibits the binding of p53 to the promoter region of the lncRNA Neat1 gene. As a result, the recruitment of SIRT1 by p53 is also disrupted, preventing its binding to the DNA. This process leads to the enrichment of H3K27ac. Subsequently, the facilitated transcription of Neat1 by p65 promotes the activation of the inflammasome.

**Figure 7 ijms-24-11032-f007:**
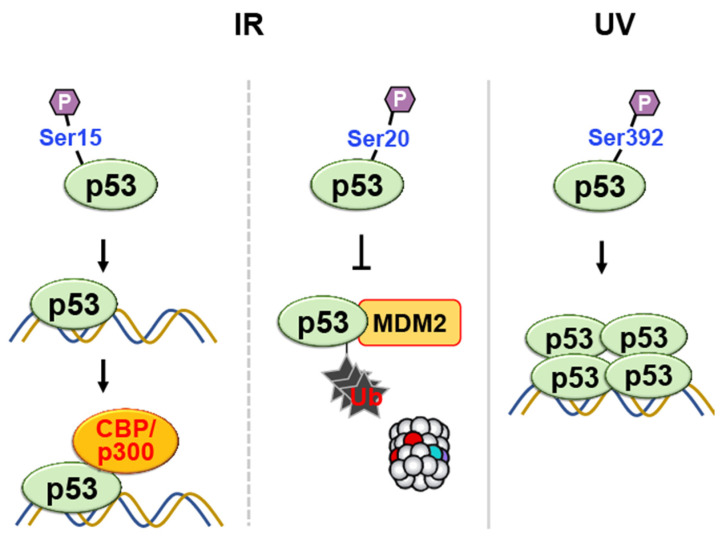
Phosphorylation of p53 at multiple sites in response to DNA damage regulates its transcriptional activity, DNA binding affinity, and protection against degradation. Phosphorylation at Ser15 and Ser20 enhances transcriptional activity and prevents ubiquitin-mediated degradation, respectively. Phosphorylation at Ser-392 enhances sequence-specific DNA binding and stabilizes tetramer formation following UV irradiation.

**Figure 8 ijms-24-11032-f008:**
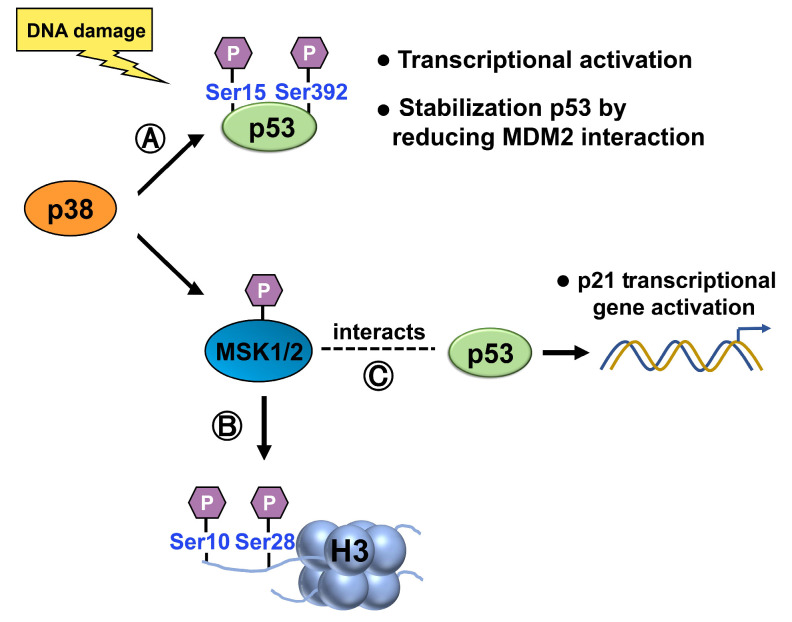
(**A**) p38 phosphorylates p53, at Ser15 and Ser392, activating p53 and increasing its stability, transcriptional activity, and induction of target genes involved in cell cycle arrest, DNA repair, and apoptosis. Ser15 phosphorylation also stabilizes p53 by reducing its interaction with MDM2. (**B**) Upon activation by p38, 1/2 phosphorylates histone H3 at Ser10 and Ser28, modulating chromatin structure and gene expression. (**C**) When activated by the p38 MAPK pathway, MSK1 interacts with p53 and is recruited to the p21 promoter, where it phosphorylates histone H3 in a p53-dependent manner.

**Figure 9 ijms-24-11032-f009:**
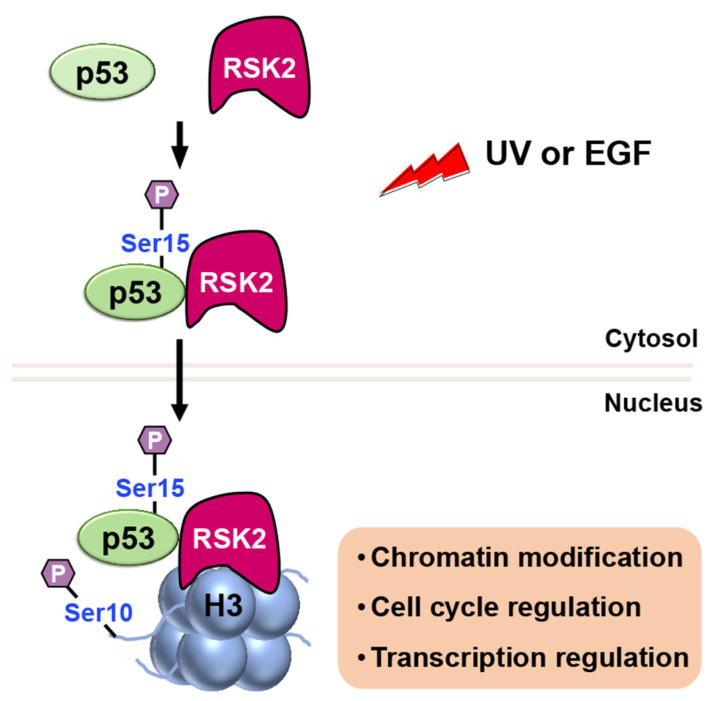
Upon stimulation with UV or EGF, activated RSK2 phosphorylates p53 at Ser15, and the RSK2/p53 complex translocates to the nucleus, where RSK2 phosphorylates histone H3 at Ser10, contributing to transcriptional regulation, chromatin remodeling, and cell cycle control.

**Figure 10 ijms-24-11032-f010:**
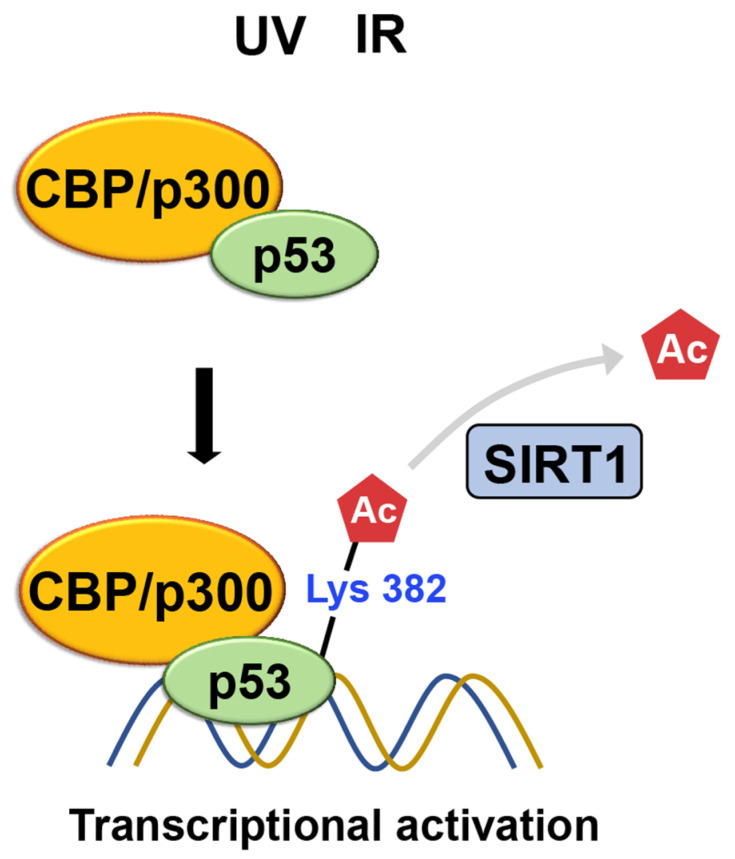
Following DNA damage caused by irradiation, lysine residues of p53 undergo acetylation. This acetylation leads to the recruitment of coactivators such as CBP/p300 and TRRAP to the p21 promoter, thereby increasing histone acetylation. Meanwhile, SIRT1-mediated K382 deacetylation inhibits p53’s transcriptional activation, promoting its degradation.

**Figure 11 ijms-24-11032-f011:**
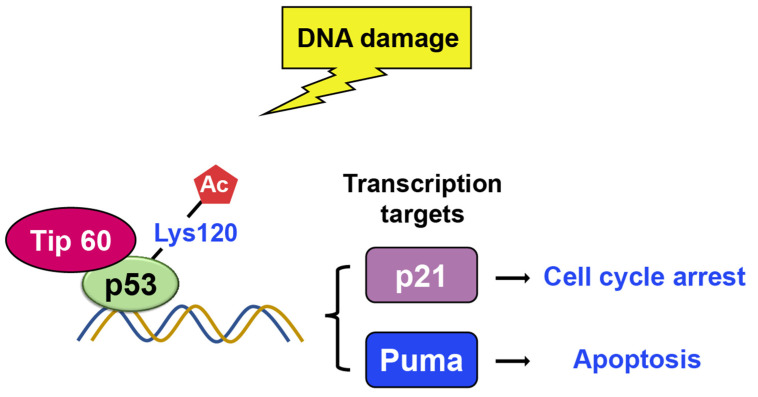
Upon DNA damage, Tip60 interacts with p53 and binds to its target gene promoters, leading to p21 activation and growth arrest. Additionally, Tip60 induces K120 acetylation, resulting in the activation of PUMA expression.

## Data Availability

Not applicable.

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
