# Peer review of "Experimental Insights into the Interplay between Histone Modifiers and p53 in Regulating Gene Expression"

_ijms, 2023, doi:10.3390/ijms241311032_

Round 1
Reviewer 1 Report
In general it is a nicely written review on TP53 and histone modifictation.
However there are some general weaknesses in which a rewrite could make this a stronger review. Although the authors correctly mention post translational modifications on TP53 which affect transcriptional activity on promoters such as p21, this section is expanded on without much reference to the underlying chromatin modifications. It may be more advantageous to go in-depth into chromatin modification studies on the TP53 gene as well as effects on histone moifications and TP53's downstream targets. This is done in the sections concerning MAPK, RSK, and G9a. In addition there is no mention on studies using acetylation inhibitors activating tumor supressor transcription, as seen in some recent studies, in addition to the chromatin remodeling with respect to H2B histone interactome. This would be a good discussion to add.
There also is a serious lack of referenced material throughout the text, especially in the introduction as well as some section from lines 373 to 487. Please go through and reference all studies and observations.
In addition the authors state they would talk about these modifications and TP53 in the context of cancer. Therefore there should be a section just on this and their proposed elucidation of new potential targets.
A major concern is in lines 62 and 70 within the figure legends. You state you recreated from a source. Did you modify? did you get author permission? You need to communicate with the editor. Perhaps a figure generated by the authors on an overall summary of effects of chromatin modifications of regulation of TP53 gene and transcriptional target activity is warranted.
Some minor edits:
lines 605 - end: are these duplicated reference list?
line 9: p53 (and if referring to protein TP53) has been investigated in the context of cancer research? TP53 is investigated in the context of a cellular process like gene transcription or apoptosis or cell cycle progression. Please reword
line 13: not This study but This review
line 34: expand what modifiers you are talking about. This is the introduction so you want to at least list them like methylation, acetylation ...
line 35: collaboration or corraboration or synergy between the two
line 37: comman after genome
line 35-50: needs references. The whole introduction needs more references
line 82: after roles put a colon ... roles: writers, erasers, readers, and movers.
Also what is the functional implications of each of these roles? Please elucidate.
line 189 spell out PRMT the first time you use an acronym
line 223 put JMJD2 in parenthesis after you spell out Jumonji C domain containing histone demethylase 2
line 259 reference?
line 262 is the EZH2 you are referring to the mouse or human. Human is capitalized while mouse gene is first letter capitalized. If human correct and correct in figure 6
lines 337- 340 references?
line 358 reference?
line 420 can take out "involved in acetylation activity' as redundant
line 441 M in Mutatin is capitalized.. lower case m please
english language is fine
Reviewer 2 Report
the Authors present a good overview of the interplay between histone modifiers and p53 in regulating gene expression.
- abstract, this is not a study. please reconcile.
- I suggest to end the Introduction at line 42 - and name the other pieces of information as a new paragraph.
- paragraph #2. can the definitions of the terms reported in bold be summarized in a Table? that would be very useful for educational (and citation) purposes.
- did the Authors follow any specific strategy for the search of papers for inclusion? if so, this should be stated (and also, it should be stated if not).
- are all the Figures necessary?
- the Perspective section can be expanded.
- Authors' contributions: should XYZ be changed into XZ? (apologies if not)
- why are references repeated twice, and with some modifications in the list?
Author Response
Reviewer 2
the Authors present a good overview of the interplay between histone modifiers and p53 in regulating gene expression.
We would like to express our sincere gratitude for your valuable feedback and insightful comments on our manuscript. Your expertise and efforts in reviewing our work are greatly appreciated. Based on your suggestions, we have made several important revisions to enhance the clarity and comprehensiveness of the manuscript.
- abstract, this is not a study. please reconcile.
Thank you very much, and we agree with the suggestion and made changes accordingly.
Line 11 This review
- I suggest to end the Introduction at line 42 - and name the other pieces of information as a new paragraph.
Thank you very much, and we agree with the suggestion and made changes accordingly.
Line 52 2. The Interplay Between Histone Modifiers and p53
- paragraph #2. can the definitions of the terms reported in bold be summarized in a Table? that would be very useful for educational (and citation) purposes.
Thank you very much for your constructive suggestion. We have created a new table per the request.
Line 139 Table 1. Histone modifiers in chromatin.
- did the Authors follow any specific strategy for the search of papers for inclusion? if so, this should be stated (and also, it should be stated if not).
Thank you for this suggestion, and we believe that these modifications strengthen our manuscript. We followed a systematic approach to ensure a comprehensive literature review. We conducted searches in PubMed using relevant keywords related to our research topic. We also reviewed the reference lists of relevant articles to identify additional studies. To minimize any potential bias, we aimed to include papers published in peer-reviewed journals up until May 2023. We have now included this information in the revised manuscript to provide transparency regarding our literature search strategy.
Now it reads,
Line 46 We implemented a systematic methodology to ensure a thorough literature review. Our approach involved searching PubMed database using selected keywords pertinent to our research topic. Additionally, we scrutinized the reference lists of relevant articles to uncover any additional studies of relevance. To mitigate potential bias, we strived to encompass papers published in peer-reviewed journals until May 2023.
- are all the Figures necessary?
We appreciate the reviewer's feedback and their consideration of the necessity of the figures in our manuscript. We believe that the inclusion of multiple figures enhances the comprehensiveness and clarity of our findings. Each figure in our manuscript serves a specific purpose in presenting key data, visualizing experimental results, and illustrating complex relationships. These figures provide essential information that supports our conclusions and aids in the understanding of the research presented. However, we understand the concern regarding the length and readability of the manuscript. To address this, we have carefully revised the figure captions and accompanying text to ensure concise and focused explanations. We believe this will help streamline the presentation while maintaining the necessary information and contextual variations. We hope the reviewer understands our rationale for including all the figures and finds that the revised manuscript conveys our research findings more effectively.
- the Perspective section can be expanded.
We agree with the reviewer. We have expanded the perspective and described potential Epigenetic Therapies Targeting Histone Modifiers and p53.
Line 538-553 Dysregulation of histone modifiers and p53 pathways can contribute to treatment resistance in cancer cells. This involvement of histone modifiers and p53 in treatment resistance has prompted the development of multiple epigenetic anticancer drugs approved by regulatory authorities. Epigenetic therapies have emerged as a promising avenue for cancer treatment. Notably, the U.S. Food and Drug Administration (FDA) has granted approval for drugs targeting diverse epigenetic categories, including HDAC, DNMT and EZH2 (Table S1). HDAC inhibitors can modulate gene expression by increasing histone acetylation levels and promoting the activation of tumor suppressor genes [97-100]. Acetylation inhibitors can activate tumor suppressor transcription. By inhibiting the activity of histone deacetylases (HDACs), acetylation inhibitors increase histone acetylation levels, leading to a more open chromatin structure. This chromatin remodeling allows easier access of transcription factors, including those involved in tumor suppressor pathways, to the target DNA sequences. With the increased accessibility, tumor sup-pressor genes such as p53 can be effectively transcribed and translated into functional proteins. Therefore, the activation of tumor suppressor transcription by acetylation inhibitors contributes to suppressing tumor progression.
- Authors' contributions: should XYZ be changed into XZ? (apologies if not)
We agree with the reviewer.
Line 576 XZ.
- why are references repeated twice, and with some modifications in the list?
Thank you very much for pointing this out. We have corrected the mistake.
Reviewer 3 Report
Specific comments to the authors
The submitted review “The Interplay between Histone Modifiers and p53 in Regulating Gene Expression” gathers, summarize and analyses heterogeneous aspects of the interaction of histone modification system and p53 on gene expression based on already published reviews as well as in-vitro, in-vivo and in-situ experiments.
First, the authors give a common introduction of histone modifiers. Further on, the authors explained in detail the possible interactions of histone modifiers and p53 in relation to methylation (via SET7/9, G9a, PRMTs, JMJD2, EZH2) and phosphorylation (via MAP Kinase cascade, p38 MAP kinase, RSK2, CBP/p300, TIP60). Finally, the authors gives a short perspective, how the interactions of histone modifiers and p53 could be relevant for cancer therapy in the future. In summary, the review gives an interesting survey of the complexity of histone modifiers and p53-axis interaction in human, which is mostly easy to read, to follow and to understand, overall. Some parts of the review should completed by the author to round out the chosen topic.
The presented topics range from histone modifiers to important interactions of histone modifiers and p53 in relation to methylation (via SET7/9, G9a, PRMTs, JMJD2, EZH2) and phosphorylation (via MAP Kinase cascade, p38 MAP kinase, RSK2, CBP/p300, TIP60). In summary, the review gives an interesting survey of the complexity of histone modifiers and p53-axis interaction in human, which is mostly easy to read, to follow and to understand, overall. The authors should add some aspects before accepting the manuscript for publication as mentioned below.
# Title: The title clarify that the presented knowledge of the “interplay between histone modifiers and p53 in regulating gene” based on experimental studies.
# Chapter “2. Methylation and 3. Phosphorylation”: Please add a table with known drugs, which could be used to targeting these histone modifiers in relation to p53.
# Chapter “3. Perspectives”: Please add ideas for possible milestones to transfer the presented findings from the bench to the bed for possible drug development in the future.
Minor editing of English language required.
Author Response
Reviewer 3
The submitted review “The Interplay between Histone Modifiers and p53 in Regulating Gene Expression” gathers, summarize and analyses heterogeneous aspects of the interaction of histone modification system and p53 on gene expression based on already published reviews as well as in-vitro, in-vivo and in-situ experiments.
First, the authors give a common introduction of histone modifiers. Further on, the authors explained in detail the possible interactions of histone modifiers and p53 in relation to methylation (via SET7/9, G9a, PRMTs, JMJD2, EZH2) and phosphorylation (via MAP Kinase cascade, p38 MAP kinase, RSK2, CBP/p300, TIP60). Finally, the authors gives a short perspective, how the interactions of histone modifiers and p53 could be relevant for cancer therapy in the future. In summary, the review gives an interesting survey of the complexity of histone modifiers and p53-axis interaction in human, which is mostly easy to read, to follow and to understand, overall. Some parts of the review should completed by the author to round out the chosen topic.
The presented topics range from histone modifiers to important interactions of histone modifiers and p53 in relation to methylation (via SET7/9, G9a, PRMTs, JMJD2, EZH2) and phosphorylation (via MAP Kinase cascade, p38 MAP kinase, RSK2, CBP/p300, TIP60). In summary, the review gives an interesting survey of the complexity of histone modifiers and p53-axis interaction in human, which is mostly easy to read, to follow and to understand, overall. The authors should add some aspects before accepting the manuscript for publication as mentioned below.
# Title: The title clarify that the presented knowledge of the “interplay between histone modifiers and p53 in regulating gene” based on experimental studies.
We would like to express our sincere gratitude for your valuable feedback and insightful comments on our manuscript. Your expertise and efforts in reviewing our work are greatly appreciated. Based on your suggestions, we have made several important revisions to enhance the clarity and comprehensiveness of the manuscript.
To address the reviewer's comment and provide a clearer indication that the presented knowledge is based on experimental studies, we can consider revising the title as follows:
Line 2 Experimental Insights into the Interplay between Histone Modifiers and p53 in Regulating Gene Expression
This revised title explicitly highlights that the information presented in the manuscript is derived from experimental studies.
# Chapter “2. Methylation and 3. Phosphorylation”: Please add a table with known drugs, which could be used to targeting these histone modifiers in relation to p53.
We have added a supplementary table that includes information about known drugs that can be used to target these specific histone modifiers in relation to p53.
Table S1. Multiple drugs within the DNMT, HDAC, EZH2 and IDH1/2 inhibitor categories have been approved by regulatory authorities, including the US FDA and other government bodies, for the treatment of diverse cancer types.
# Chapter “3. Perspectives”: Please add ideas for possible milestones to transfer the presented findings from the bench to the bed for possible drug development in the future.
We have expanded the perspective and described potential Epigenetic Therapies Targeting Histone Modifiers and p53.
Line 538-553 Dysregulation of histone modifiers and p53 pathways can contribute to treatment resistance in cancer cells. This involvement of histone modifiers and p53 in treatment resistance has prompted the development of multiple epigenetic anticancer drugs approved by regulatory authorities. Epigenetic therapies have emerged as a promising avenue for cancer treatment. Notably, the U.S. Food and Drug Administration (FDA) has granted approval for drugs targeting diverse epigenetic categories, including HDAC, DNMT and EZH2 (Table S1). HDAC inhibitors can modulate gene expression by increasing histone acetylation levels and promoting the activation of tumor suppressor genes [97-100]. Acetylation inhibitors can activate tumor suppressor transcription. By inhibiting the activity of histone deacetylases (HDACs), acetylation inhibitors increase histone acetylation levels, leading to a more open chromatin structure. This chromatin remodeling allows easier access of transcription factors, including those involved in tumor suppressor pathways, to the target DNA sequences. With the increased accessibility, tumor sup-pressor genes such as p53 can be effectively transcribed and translated into functional proteins. Therefore, the activation of tumor suppressor transcription by acetylation inhibitors contributes to suppressing tumor progression.
Round 2
Reviewer 1 Report
The authors include a very nice section on therapeutic modulators and the figures are very descriptive and well thought out. This will be a very well received review.